# Current Insights on Vegetative Insecticidal Proteins (Vip) as Next Generation Pest Killers

**DOI:** 10.3390/toxins12080522

**Published:** 2020-08-14

**Authors:** Tahira Syed, Muhammad Askari, Zhigang Meng, Yanyan Li, Muhammad Ali Abid, Yunxiao Wei, Sandui Guo, Chengzhen Liang, Rui Zhang

**Affiliations:** Biotechnology Research Institute, Chinese Academy of Agricultural Sciences, Beijing 100081, China; syedtahira98@gmail.com (T.S.); 2017Y90100082@caas.cn (M.A.); mengzhigang@caas.cn (Z.M.); liyanyan01@caas.cn (Y.L.); abid@caas.cn (M.A.A.); weiyunxiao@caas.cn (Y.W.); guosandui@caas.cn (S.G.)

**Keywords:** *Bacillus thuringiensis*, vegetative insecticidal proteins, insecticidal activity, resistance, pyramids

## Abstract

*Bacillus thuringiensis* (Bt) is a Gram negative soil bacterium. This bacterium secretes various proteins during different growth phases with an insecticidal potential against many economically important crop pests. One of the important families of Bt proteins is vegetative insecticidal proteins (Vip), which are secreted into the growth medium during vegetative growth. There are three subfamilies of Vip proteins. Vip1 and Vip2 heterodimer toxins have an insecticidal activity against many Coleopteran and Hemipteran pests. Vip3, the most extensively studied family of Vip toxins, is effective against Lepidopteron. Vip proteins do not share homology in sequence and binding sites with Cry proteins, but share similarities at some points in their mechanism of action. Vip3 proteins are expressed as pyramids alongside Cry proteins in crops like maize and cotton, so as to control resistant pests and delay the evolution of resistance. Biotechnological- and in silico-based analyses are promising for the generation of mutant Vip proteins with an enhanced insecticidal activity and broader spectrum of target insects.

## 1. Introduction

Common soil bacterium *Bacillus thuringiensis* (Bt) is a Gram negative motile bacterium that gained significant popularity over the past decades because of its role as an invertebrate pest killer [1]. *B. thuringiensis* has been extensively studied for its ability to produce immense arsenal of toxins with an insecticidal potential against insect vectors of human diseases, agricultural pests, nematodes, fungi, gastropods, and protozoans [1,2,3,4,5]. Bt proteins, as an active ingredient of biopesticides, are a valuable eco-friendly approach to replacing chemical insecticides. Bt genes are engineered in many crops like maize, cotton, soybean, and rice, offering a sustainable solution to control insect pests [6]. A Bt-transformed cotton plant expressing Cry1Ac was first introduced in 1996 in Australia and the United States of America [7]. The agriculture land covered by transgenic Bt crops reached 98.5 million hectors in 2016 [8]. Nevertheless, the rise of insect resistance is becoming a major hurdle in the commercialization of Bt transgenic crops [9].

*B. thuringiensis* produce Crystal (Cry) and Cytolytic (Cyt) toxins during sporulation, which are stored in parasporal crystalline inclusions and released after the disintegration of the cell wall in a culture medium. However, vegetative cells produce non-crystalline toxins, such as vegetative insecticidal proteins (Vip) and secreted insecticidal proteins (Sip), secreted as soluble proteins in a growth medium. Vip proteinaceous toxins were isolated from the culture medium of both *Bascillus cereus* and *B. thuringiensis* after screening [10,11]. The Vip toxin family is classified into four subfamilies, namely Vip1, Vip2, Vip3, and Vip4, based on their amino acid similarity, which also guides Vip proteins’ nomenclature [10,12]. The Bt Toxin Nomenclature Committee assigned a four-rank name to each toxin—primary rank (two toxins with 45% similarity), denoted by an Arabic number; secondary rank (<78% similarity), denoted by an uppercase letter; tertiary rank (95% similarity), denoted by a lower case letter; and quaternary rank (˃95% similarity), denoted by the final number (Figure 1). 

To date, 15 Vip1 proteins, 20 Vip2 proteins, 111 Vip3 proteins, and 5 Vip4 proteins have been reported and named [13]. (Vip1 and Vip2 heterodimer toxins are effective against insects from Coleopteran and Hemipteran orders [14]. The largest family, Vip3, is effective against many species of Lepidoptera, and crops like cotton and maize have been successfully transformed with various Vip3 toxins [15]. Interestingly, Vip proteins have no sequence homology with Cry proteins, and do not share common binding sites in target insects [16,17,18,19]. This makes them ideal toxins to be used in combination with Cry proteins in insect resistance management (IRM) programs. 

Significant knowledge about the structure and mode of action is available for Cry proteins, but this information is still rudimentary for Vip toxins. This review discusses recent insights on the structure and mechanism of action of Vip toxins. Detailed knowledge about the structure and functional characterization of Vip toxins will lead to the development of new strategies for designing improved toxins against insects that have developed resistance. The Vip family of Bt toxins is a potential candidate against these resistant pests. This study also focused on the natural and in vitro evolution of Vip toxins, and the strategies developed so far that improve the insecticidal activity of these toxins at a molecular level.

## 2. Structure and Function of Vip Toxins

Vip proteins are widely distributed among Bacillus species. These proteins are not produced in parasporal crystalline inclusions, but are instead secreted into the culture medium. Like other Bt toxins, Vip proteins are also inactive in their native form, and are activated after being secreted in the membranes of insect midgut cells through the action of enzymes [20]. Detailed structural information of the Vip proteins is not yet elucidated; therefore, the current structural and functional predictions are based on in silico and mutagenic studies. A comparison between the Vip family of proteins is summarized in Table 1.

### 2.1. Structure of Vip1/Vip2 Binary Toxins

Vip1 and Vip2 act as binary toxins possessing an ADP ribosyltransferase activity [21]. The genes of the Vip1 and Vip2 proteins are located in a ~4 kb single operon with different reading frames [12,22]. To date, 35 *Vip1* and *Vip2* genes have been listed in the Bt nomenclature database. Sequence analysis found that Vip1 is synthesized as a protoxin of 100 kDa, with an N terminal signal peptide sequence of 35 amino acids. Similarly, Vip2 proteins in protoxin form are 52 kDa in size, with an N terminal signal peptide of 50 amino acids [22,23]. After their modification at the N terminal signal peptide, Vip1 and Vip2 are transformed into a mature proteins of 80 kDa and 45 kDa, respectively [24]. The structural analysis of Vip2 has confirmed two domains, N-terminal (Nt) and an NAD-binding (Nicotinamide adenine dinucleotide) C-terminal (Ct) domain [15,25]. Moreover, X-ray crystallography revealed homology between the N and C terminal domains of the Vip2 protein, and both domains are formed by the perpendicular packing of five mixed β sheets, with one flanking α helix and three anti parallel β sheets with three flanking α helices [25].

Research based on their sequence homology suggests that Vip1 and Vip2 act as binary toxins of A + B type, with similarities to many mammalian toxins. Vip1 has very little structural similarity with the *Clostridium spiroforme* toxin, protective antigen of *Bacillus anthracis*, and CdtB toxin of *Clostridium dificile*. Vip2 has a structural similarity to the active domain of CdtA toxin produced by *C. difcile* [26]. Both Vip1 and Vip2 toxins have a similarity to the C2 toxin of *Clostridium botulinum* and the domain Ia of the iota toxin produced by *Clostridium perfringens* [27]. Altogether, this predicts Vip2 as an ADP-ribosyltransferase toxin inhibiting the polymerization of actin filaments, causing cytoskeleton abnormalities and insect cell death [28]. Vip1 is inferred to act as a B toxin (Binding domain) responsible for the translocation of Vip2 inside insect midgut cells. Vip2 is a cytotoxic A toxin with a binary toxin response, showing no toxicity to insects when applied alone [28].

### 2.2. Structure and Function of Vip3 Proteins

Vip3 toxin is also produced during the vegetative growth phase of Bt, and Vip3A is the most widely studied Vip toxin so far. Vip3 is a diverse family of toxins with 95% similarity between its members, and it shares no primary sequence homology to any other Vip families or Bt toxins [29]. These proteins show a strong inhibition of insect larval growth at a low concentration [29,30], and the structural differences among the Vip3 members predict a broader mode of action against a wide spectrum of insects.

The Vip3 gene encodes a protein of 89 kDa having 787 to 789 amino acids [29]. Functional characterization of the *vip3Aa16* gene revealed that its −35 and −10 promoter regions have homology to the *Bacillus subtilis* promoters, which suggests that the *Vip3* gene promoter is under the control of the σ^35^ holoenzyme [31]. It shares no sequence homology with any other toxins produced by *B. thuringiensis* [1]. Phylogenetic analysis showed that the Vip3 protein belongs to distant clade rather than to Cry toxins. 

A comprehensive structure of Vip3A has not yet been resolved, and is only derived through in silico modeling [32,33]. Notably, the Vip3A signal peptide located at the N-terminal is responsible for the translocation of protein. The N-terminal region is highly conserved and performs important regulatory insecticidal functions [34], while the C-terminal region undergoes various modifications and has the ability to target insect specificity. At present, the role of both domains in insecticidal activity is mainly perceived by mutagenic studies. The deletion or addition of amino acids at the N- and C-terminals greatly affects the entomocidal property of a particular protein. For example, the deletion of amino acids at the Nt region causes a negative effect on the insecticidal activity of the Vip3A protein [35,36]. Substitutions at position T167 or G168 at the N-terminal of Vip3Af with alanine leads to a decreased insecticidal activity [32]. In contrast, the deletion of 200 amino acids from the N-terminal enhanced the toxicity of Vip3Aa against various Lepidoptera pests, and the deletion of 200 amino acids from the C-terminal region abolished the insecticidal activity of Vip3BR. Also, the deletion of 127 amino acids at the C-terminal maintained a low level of insecticidal activity against *Agrotis ipsilon* (Lepidoptera: Noctuidae) and *Helicoverpa armigera* (Lepidoptera: Noctuidae) [37], which suggests the important roles of the N- and C-terminal parts in the insecticidal activity of Vip3 proteins. It is also possible that the interactions of other plant or insect proteins with Vip3 C- and N-terminal domains could enhance their insecticidal activity. Hence, further studies are needed to explore the exact mechanism of action.

Each C-terminal amino acid plays an important role in the target specificity and toxicity against many Lepidopteron pests. Meanwhile, Vip3A11 mutants are generated after replacing nine residues at the C-terminus with Vip3A39 residues by site-targeted mutagenesis. Here, the cysteine residue CYS784 of the C-terminal region is found to be a crucial trypsin cleave site for bioactivity and toxicity [38]. However, the Vip3 C-terminal region alone does not possess an insecticidal activity, as the expression and purification of the C-terminal region of Vip3Ab1 and Vip3Bc1 cause no harm to the insects. Therefore, in contrast with the Cry proteins, both the C- and N-terminal regions are important for oligomerization and proteolytic stability, with a significant contribution to the toxicity of the Vip3 proteins [39].

Site directed mutagenesis anticipates putative trypsin cleave sites Lys195, Lys197, and Lys198 inside Vip3Aa. The mutants generated by replacing these three Lysine residues with alanine lose sensitivity to trypsin or midgut juices (MJ), and also show toxicity against *Spodoptera exigua* (Lepidoptera: Noctuidae) [40]. In the same manner, the substitution of cysteine with serine at the C-terminal also reduces the Vip3A7 protein insecticidal activity against *Plutella xylostella* (Lepidoptera: Plutellidae), possibly due to the disruption of disulfide bonds between cysteine residues [41]. An alanine scanning analysis of 588 residues unveiled a five-domain structure of the Vip3Af1 protein and its role in toxicity. This approach revealed 50 residues with a significant impact on Vip3Aa structural conformation and toxicity. Among them, two clusters of 19 substitutions, located near the N-terminus region between Leu167–Tyr27 or on the C-terminus between Gly689–Phe741, abolished toxicity to *Agrotis segetum* (Lepidoptera: Noctuidae). Another 19 substitutions also reduced toxicity to *Spodoptera frugiperda* (Lepidoptera: Noctuidae). Hence, it is evident that each amino acid within the Vip3 protein plays a diverse role in protein stability and toxicity [32]. Transmission electron microscopy and single particle analysis of Vip3Ag4 have revealed the surface topology of its tetramers. After trypsin treatment, the protein forms an octamer containing tetramers of 65 kDa and 22 kDa fragments. In addition, the tetrameric form and main topology are retained even after trypsin treatment [42]. 

Although no clear evidence about the Vip3 3D structure is available, in silico analyses point to the presence of five domains in the Vip3Af protein. The trypsin fragmentation of alanine mutants depicted five domains in the Vip3Af proteins structure. Domain I spans from amino acids 12 to 198, domain II 199 to 313, domain III 314 to 526, domain IV 527 to 668, and domain V 669 to 788 amino acids. Domains I to III are necessary for tetramerization, however not domain V. In addition, the role of domain IV remains unclear [43]. This evidence is further supported by the 2D structure of the Vip3B protein predicted by X-ray diffraction studies. According to this model, Vip3B is composed of five domains; two domains carrying α helices (DI and DII) at the N-terminus and three β sheet containing domains (DIII, DIV, and DV) on the C-terminal region. Domain III shows a slight resemblance to domain II of the Cry4Aa and Cry1Ac proteins. Domain IV and V share homology with carbohydrate binding modules (CBM), indicative of glycosylated receptor-binding inside the midgut [44]. A carbohydrate binding motif (CBM; CBM_4_9) has also been identified at the C-terminal region in all of the Vip3 protein members, except for Vip3Ba. It is then possible that the C-terminal region plays a crucial role in recognition and binding to midgut receptors [45]. 

A recent report sheds more light on the structure of all five domains of Vip3A and their related function. Domain II has two highly conserved hydrophobic α helices, predicting their role in membrane insertion and pore formation inside the insect midgut. Domain III comprises three β sheets potent for cell binding, along with domain II, persistent with previous results. In Vip3A, two CBM domains with different glycan binding pockets are found in C-terminal region, which forms domain IV and domain V [46]. This indicates a specificity in their binding capacity with glycan on the targeted cell surface. The cryo-EM (cryogenic electron microscopy) structural analysis solves the structure of Vip3A and showed how the toxin forms pores in the insect midgut. The protein architecture has five distinct domains in protoxin form, with domains I (coiled α 1–α 4) ending at the primary protease cleavage site, and domain II having five α helices mainly producing the core before trypsin digestion. Antiparallel β sheets of domain III form a β prism fold analogous to the Cry toxins. In the Cry toxin, this fold assists in receptor recognition. Similar to previous results, two CBM folds form the last two domains. After trypsinization, all four monomers stay connected, and no conformational change takes place in domains II to V. Three N-terminal α helices form a parallel four helix coiled coil, which forms a long dipole to lodge the ions in its cavity. Its dimension has the ability to form pores in the lipid bilayer [47].

### 2.3. Vip4 Toxins

Vip4 is the least characterized toxin of the Vpb class. Only five Vip4 proteins have been identified to date. The first reported Vip4 toxin was Vip4Aa1 (now named Vpb4Aa1), isolated from Bt strain Sbt009, with no insecticidal activity against any pests. Its molecular mass is ~108 kDa and it possesses 965 amino acids [13]. The main region spanning from 47 to 77 amino acids is a PA14 domain, and the region from 218–631 residues is named the bacterial Binary_ToxB domain. This novel protein shares 34% identity with the Vip1Aa1 protein and 65% with the Ia domain of the Iota toxin of *B. cereus*, specifically to the B component of the binary toxin [26].

## 3. Modern Classification of Vip Proteins

Recently, the classification and names of the Vip family of toxins were modified by Crickmore et al. [48]. Accordingly, all Vip toxins are placed in three different classes, namely Vip3, Vpa, and Vpb. The Vip3 family mnemonic remains the same. In class Vpa, all of the Vip2 proteins are placed. Class Vpb contains a binary toxin component of Vip2, such as Vip1, and its structural analogues, previously known as Vip4 [13].

## 4. Mechanism of Action

The mode of action of the Vip toxins inside insect guts is not yet clear and needs further investigation. Differences in the structure of the Vip and Cry toxins determine the different target sites in the insect midgut, making them suitable candidates for insects-resistant control [16]. 

### 4.1. Mechanism of Action of Vip1/Vip2 Binary Toxin

There is no clear mechanism of action for the Vip1/Vip2 binary toxin. Each protein alone is not enough to cause toxicity, but rather they act in combination as A + B binary toxins [28]. This multistep process begins with toxin entry at the insect midgut and ends with larvae death. There are many proposed models on this mechanism. Toxins first enter the midgut and are digested by trypsin-like proteases. Enzymatic action by trypsin or midgut juices (MJ) cleaves Vip1Ac into its activated form before entering into the brush border membrane (BBM). After enzymatic activation, Vip1Ac oligomerizes to form seven monomers containing a multimeric structure [49], which binds the BBMVs of cotton aphids with a high target specificity [14]. Similar results were observed with the activated form of the Vip1Ad and Vip2Ag binary toxin activity. Vip2Ag binding, insecticidal activity, and toxicity is increased tremendously in the presence of Vip1Ad in *Holotrichia parallela* (Coleoptera: Scarabaeidae). This shows strong evidence for Vip1 as a potential receptor of Vip2 after trypsin activation [50]. 

After internalization, Vip2 transfers the ADP ribose group to the actin filaments, inhibiting its polymerization. This ultimately leads to abnormal microfilament formation and cell death [15,23]. The insecticidal activity of Vip2Ag and Vip1Ad is characterized by the vacuolization and destruction of BBMVs and microvilli depletion, similar to what has been observed in *H. armigera* fed on Cry2Ab and Vip3AcAa/Cry1Ac binary toxins [51,52]. In a nutshell, the histopathological effects of Vip1/Vip2 binary toxins are comparable to those of Cry proteins.

### 4.2. Mechanism of Action of Vip3 Toxin

The Vip3 toxins mechanism of action has some similarity to the Cry toxin-like protease activation, binding with midgut cells and pore formation. The complex multistep process is not fully elucidated yet. At first, all Vip3 toxins are activated by midgut juices. Proteolytic analysis of Vip3A toxin has revealed several fragments (62–66 kDa, 45 kDa, 33 kDa, and 19–22 kDa) with a similar pattern after trypsin treatment or insect midgut proteases. The main cleavage product from the C-terminus region is a 62–66 kDa core peptide, generally considered to be the main part of the activated toxin [19,37,53,54]. However, a 19–22 kDa fragment comprising the 199 amino acids at the N-terminus, together with a 62–66 kDa fragment, are crucial for lethality, as shown by bioactivity assays after purification [43]. Furthermore, 45 kDa and 33 kDa fragments are formed after cleavage of a 62–66 kDa fragment [53]. Other research on protease cleavage inferred that these smaller fragments formed after the digestion of 65 kDa fragment could be the result of denaturing conditions of SDS-Page, as the C-terminal (65 kDa fragment) domain remains intact under native conditions [55]. In contrast, the proteolytic digestion of Vip3Ca produces a 70 kDa fragment [56]. Variations in the insecticidal activity of the Vip3 toxin seem to depend on the hydrolysis pattern inside insects’ midgut [54], however, the proteolytic digestion pattern of Vip3Ab1 and Vip3Bc1 in a non-susceptible insect, *Ostrinia nubilalis* (Lepidoptera: Crambidae), was identical to the susceptible insects, and shed light on the fact that resistance might be unrelated to protein cleavage [39]. 

After the trypsin processing of Vip3A, two of its fragments, 19–22 kDa of the N-terminal region and 62–66 kDa fragment of the C-terminal region join to form a ~360 kDa homo-tetramer, which cannot be degraded by proteases [57]. Moreover, another study proposed the formation of a >240 kDa complex and identified a novel site, S164, crucial for the formation and stability of this complex. Mutations at this site lead to a loss of insecticidal activity [58]. In addition, the two fragments are eluted together in gel permeation chromatography, emphasizing the fact that they may remain together after cleavage [55]. The interaction between the 22 kDa and 65 kDa fragment is necessary for their stability and toxicity [39].

The next phase is the binding of the activated toxin with the BBMVs inside the midgut of susceptible insects. The Vip3 protein binding sites do not overlap with the Cry proteins. A clear mechanism regarding the binding of the Vip3 toxin is still not available, and only a few studies have addressed the recognition of binding molecules inside the insect midgut cells. So far, only a few Vip3 binding proteins have been identified, one of which is ribosome S2 protein, identified by yeast hybrid assay, and confirmed by vitro pull-down assays in sf21 cells of *S. frugiperda*. A knock down of the ribosomal S2 gene in the Sf21 cells and the larvae of *Spodoptera litura* (Lepidoptera: Noctuidae) resulted in a reduced larvicidal activity, considering that S2 is one of the proteins involved in the Vip3 insecticidal mechanism [59]. Another potential protein is a 48 kDa tenascin-like glycoprotein, which strongly binds to Vip3Aa in BBMVs from black cutworm [36,60].

Many current studies are focused on the recognition of Vip3 potent receptors in order to understand the mechanism of action. A novel receptor, fibroblast growth factor receptor-like protein (Sf-FGFR), has been identified on the membrane of Sf9 cells, as a result of its binding affinity toVip3, confirmed by in vitro analysis. Silencing of the *Sf-FGFR* gene resulted in a reduced toxicity of Vip3Aa to Sf9 cells. The localization of Sf-FGFR and Vip3Aa on the surface and then inside the cytoplasm suggests that binding takes place on the surface, leading to internalization [61]. Similarly, Vip3Aa has shown a strong interaction with scavenger receptor class C like protein (Sf-SR-C) in both in vivo and in vitro analysis with Sf9 cells of *S. frugiperda*. Knocking down the expression of these receptor genes results in a reduced mortality of Sf9 cells and *S. exigua* larvae to Vip3Aa [62]. Further studies are required to fully clarify the specificities of the Vip3 toxin receptor binding. 

Proteolytic activation and receptor mediated binding of Vip3 toxins leads to pore formation and cell death. After feeding Vip3 toxins, the insect midgut is damaged, which is proposed as the main target site for the Vip3 toxin. Histopathological analyses have revealed similar symptoms to Cry toxins, like swollen or lysed midguts and pore formation [19,63]. However, clear mechanisms on toxin intercellular localization and pore formation in BBM are not available. The localization mechanism of active the Vip3Aa protein inside Sf9 cells, by laser scanning confocal microscopy with fluorescently labeled Vip3Aa (Alexa488-actVip3Aa), has demonstrated that Vip3Aa is not internalized by the endocytic- or clathrin-dependent endocytic pathway. Instead, this seems to happen through receptor mediated endocytosis, after which the Vip3Aa protein interacts with various cytosolic proteins (e.g., ribosomal S2 protein) [64].

Voltage clamping and planar lipid bilayer experiments predict the ability of a toxin to form discrete ion channels without involving receptors, which differs from Cry1Ab [16]. Pore formation in lipid bilayers by an activated toxin inside the midgut has been reported in H. armigera using the florescent quenching method [65]. Additionally, the maximum potential of the activated Vip3Aa toxin to form pores has been seen at specific pHs during in vitro analyses, showing that pore formation only happens at acidic or neutral pH [57]. The mechanism of Vip3 pore formation and virulence is generally the most accepted (Figure 2). 

Contrary to the pore formation model, is another mechanism in which the Vip3 toxin induces apoptotic cell death in insects. The intercellular localization of Vip3 causes abnormalities of cell division and leads to the apoptosis of insect midgut cells. Vip3A treated Sf9 cells undergo arrest at the G2/M phase and the disruption of the mitochondrial membrane potential (∆Ψm), leading to apoptotic cell death via the sf-caspase-I mediated pathway [62]. Another study has evidenced the involvement of regulatory proteins and lysosomes in apoptosis. Furthermore, symptoms of apoptosis and mitochondrial collapse are prevalent in sf9 cells when administered Vip3Aa, such as the accumulation of reactive oxygen species (ROS), caspases (caspase 3,9), and cytochrome c [66] (Figure 3). Because of the lack of sound information, more investigation is needed to clarify the downstream mechanism of Vip3 induced apoptosis and cell death.

To understand the mechanism of insect response to toxins, the transcriptomic and proteomic characterization of genes and proteins is of great interest. The gene expression profiles of toxic-dose-treated larvae of *S. exigua* and *S. litura* have been analyzed in two independent studies. From the analysis of the transcriptome profile in *S. exigua* larvae, ˃29,000 unigenes were obtained, in which the S2 and tenascin-like protein gene expression, was stable. The up regulated genes were mostly related to immune reposes and defense mechanism while down regulated genes were mainly metabolic ones [67]. Similarly, the genes coding for lysosomes and antimicrobial peptides have been found to be up-regulated in *S. exigua* [68]. In the gene expression profile of the Vip3 toxin treated larvae of *S. exigua*, immune response genes are up-regulated and the genes involved in the digestion process are down regulated. The up regulation of initiator and effector caspases genes and antimicrobial effectors provides strong evidence for the apoptosis of insect cells, similar to previous reports [69]. In another analysis, 56,498 unigenes were identified in *S. litura* larvae. The transcription levels of the trypsin related genes increased in this case after toxin induction, which supports the role of trypsin in the metabolism of the Vip3Aa toxin [70]. However, further investigation is necessary to elucidate how Vip3 toxins cause apoptotic cell death. 

## 5. Insecticidal Activity of Vip Proteins 

After research on various insect species, it has been found that Vip1/Vip2 has an insecticidal activity against some pests of Coleopteran and Hemipteran orders [14]. Vip1 and Vip2 act in combination, and none of the toxins have an insecticidal activity when administered alone. The combination of Vip1Aa/Vip2Aa or Vip2Ab has been found to be affective against Diabrotica spp [12]. The Vip1Ad/Vip2Ag toxins, when combined and expressed as a binary toxin, show toxicity against *H. parallela*, *H. oblita,* and *Anomala corpulenta* (Coleoptera: Scarabaeidae) [24].

The most extensively studied Vip3 protein family is Vip3Aa, which is widely known for their insecticidal activity against many species of Lepidopteron and pests like *S. exigua*, *H. armigera*, *S. frugiperda*, *Heliothis virescens* (Lepidoptera: Noctuidae), *Helicoverpa zea* (Lepidoptera: Noctuidae), and *A. ipsilon*. Although they have very minor differences in their sequences, Vip proteins exhibit great variability in their targeted insects. For instance, one of the most recently discovered members of the Vip3 family protein, Vip3Ca, has 70% homology with Vip3Aa and has been found to be toxic against *Chrysodeixis chalcites* (Lepidoptera: Noctuidae), *Mamestra brassicae* (Lepidoptera: Noctuidae), and *Trichoplusia ni* (Lepidoptera: Noctuidae). However, the Vip3Ca toxin shows a moderate insecticidal activity against *Cydia pomonella* (Lepidoptera: Tortricidae; non-susceptible to the Cry toxin) and *O. nubilalis* (susceptible to the Cry toxin) [71]. Vip3Ca is the most potent toxin against *Mythimna separate* (Lepidoptera: Noctuidae), with an LC_50_ value 3.4 µg/g. This toxin could be used in future maize crop protection to control Oriental armyworm [72]. Vip3Ca is also more toxic to *Ostrinia furnacalis* (Lepidoptera: Crambidae), which is more similar to Cry1Ab than Vip3A, and can be an effective candidate against Cry1Ab-resistant colonies of *O. furnacalis* [73]. 

Likewise, Vip3Ae and Vip3Af are toxic to *Spodoptera littoralis* (Lepidoptera: Noctuidae), *M. brassicae,* and *Lobesia botrana* (Lepidoptera: Tortricidae), and Vip3Ab is lethal against *A. ipsilon*. Nonetheless, Vip3Ad exerts no toxicity to insects like *H. armigera, M. brassicae, S. frugiperda, S. exigua, S. littoralis,* and *A. ipsilon* [29]. The insecticidal activity of Vip3Aa59 is significantly higher than Vip3Aa58 towards *Dendrolimus pini* (Lepidoptera: Lasiocampidae) [20], and Vip3Aa45 shows a 40-fold higher toxicity than Cry proteins against *S. exigua* [26]. 

In some reports, the toxicity of the Vip3 protoxin was found to be more than active toxin, e.g., Vip3Ae protoxin is more insecticidal than the active toxin when compared with Vip3Aa [54]. These could result from differences in the protocols for protein isolation and purification, bioassay conditions, and quantification methods. For example, metal chelate chromatography clearly affects the insecticidal activity of Vip3 [20,54].

## 6. Evolution of Resistance and Cross Resistance to Vip3 Toxins

Since the application of Vip3 transformed Bt crops, only few cases of practical resistance have been reported. The reported cases of resistance are from both laboratory and field selected insects. For example, in laboratory conditions, insects of *S. litura* show a 280-fold resistant to Vip3A after 12 generations of selection. This is probably due to the lower protease activity in those insects [74]. Similarly, the laboratory selection of *H. virescens* with Vip3Aa for 12 generations results in a ˃2040-fold resistance. This resistance is polygenic and decreases after 13 to 28 generations, without toxin administration. A lack of cross resistance is also seen against Cry toxins [75]. 

In a field evolved resistance study, conducted with transgenic maize expressing Vip3Aa20 in Brazil, the target pest, *S. frugiperda,* showed a >3200-fold resistance. The pattern of resistance inheritance was autosomal recessive and monogenic, with a very low frequency of resistant alleles (0.00009 estimated by the F_2_ screening method) [76]. A Vip3A resistant strain of *S. frugiperda* evolved >632-fold resistance to Vip3A, with minor cross resistance to Cry1F, Cry2Ab2, or Cry2Ae toxins [72].

Despite the fact that Vip3A expressing crops are not yet commercialized in Australia, a high frequency of resistant alleles has been observed in various studies conducted on *H. armigera* and *Helicoverpa punctigera* (Lepidoptera: Noctuidae). In a study screening for resistance alleles in *H. armigera* and *H. punctigera* using the F_2_ method, a natural polymorphism and very high baseline frequency of 0.027 and 0.008, respectively, were observed, with no cross resistance to Cry1Ac or Cry2Ab. The presence of both resistance alleles on the same locus confirmed resistance to be recessive [77]. Another biochemical study also found a high frequency of resistant alleles in the same insect, *H. armigera,* with no significant change in the binding of Vip3 toxins to BBMVs compared with susceptible insects. Instead, a low proteolytic activity was the main driver of Vip3 resistance [78]. 

Despite the intrinsic specificities of Vip3 toxicity, many studies have observed Vip3A cross-resistance to Cry proteins. Cross resistance to Vip3C has been observed in insects of different species, previously found to be resistant to Cry1A, Cry2Ab, Dipel (Mixture of Cry1 and Cry2), and Vip3. Vip3C shows cross resistance to *H. armigera* colonies resistant to Vip3Aa or Vip3Aa/Cry2Ab, and toxicity against *O. furnacalis,* which is nonsusceptible to Vip3A [73]. The biochemical basis of resistance could be the down regulation of membrane bound alkaline phosphatase (mALP) isoform HvmALP1, observed in Vip3 resistant insects, which is thought to be the functional receptor of Vip3. In addition, mALP can be used as a marker for the detection of Vip3A resistance [79]. Moreover, Cry1F and Vip3A do not share common binding sites in *S. frugiperda* [17], and also lack cross resistance [54].

## 7. Identification of Bt Isolates Containing Vip Genes

Over the past few decades, researchers have been trying to find new *B. thuringiensis* isolates from different geographical regions and diverse environments, to develop new toxins with a high insecticidal potential and to cope with resistance. The discovery of new isolates not only helped in the production of new pesticides of a wide insecticidal spectrum, but also in overcoming insecticidal resistance [80]. *B. thuringiensis* strains were isolated from diverse habitats, like milk and mossy pine cone [81], soil, leaf [82] and insect cadavers [83], and goat gut [84]. 

After the characterization of native *B. thuringiensis* isolates isolated from soil, and fig leaves and fruits from a Turkish collection, a new *B. thuringiensis* isolate, 6A, was identified carrying a high expression of Vip3Aa. The identified protein, named Vip3Aa65, has a similar insecticidal activity against *Grapholita molesta* (Lepidoptera: Tortricidae) and *H. armigera,* but is less toxic to Spodoptera spp. compared with Vip3Aa16 [85]. In another study, two Bt strains, BnBt and MnD, were found to produce Vip proteins in isolates of great potential with a high toxicity (LC_50_ = 41.860 ng/μL of BnBt and 55.154 ng/μL of MnD) in the second instar larvae of *S. littoralis* [86]. Lone and co-workers isolated and expressed a novel Vip3Aa61 gene in *Escherichia coli* from isolate *B. thuringiensis* JK37. Nucleotide analysis found differences in many amino acids compared with Vip3A. Because of its high insecticidal activity (LC_50_ = 169.63 ng/cm^2^) against second instar larvae of *H. armigera*, the Vip3Aa61 toxin is a potential candidate for transgenic crop production and pest protection [87]. 

Bacterial 16S ribosomal RNA (rRNA) sequencing is frequently used to characterize microbes at a species rank. In the past, this method was not successful because of the close resemblance of *B. thuringiensis* to other strains, making it ambiguous for phylogenetic and diversity analysis [88]. A new and more reliable method of phylogenetic analysis is multiple locus sequence typing (MLST). MLST-based analysis of *B. thuringiensis* kurstaki isolates from Assam, India, confirmed the presence of Vip1 (53.3%), Vip2 (46.6%), and Vip3 (40%) genes [89]. The results were in contrast to a previous study, where the Vip3 gene was more abundant than Vip1 and Vip2 [90]. Recently, the characterization of an indigenous Bt strain, found Vip3 gene in this strain, and that the spore crystal mixture of this isolate had a high mortality rate against *S. frugiperda* [91].

## 8. Transgenic Crops Expressing Vip Proteins

Despite the high toxicity of Vip1/Vip2 toxins against corn rootworms, Bt maize crops containing these binary toxins cannot be developed because of the cytotoxicity of Vip2 proteins [78].

Vip3 proteins are introduced in crops like cotton and maize. The *Vip3Aa19* gene was first introduced into Bollgord cotton (COT102), expressed as a single insecticidal protein (VIPCOT commercialized in 2008 in Unite States of America). This toxin provided protection against three major cotton pests, *H. virescens*, cotton bollworm *H. zea*, and *Pectinophora gossypiella* (Lepidoptera: Gelechiidae). Later, it was pyramided with Cry genes (modified Cry1Ab) for insect resistance management. Vip3Aa20 was introduced (MIR162) in corn and was commercialized in 2009 in the United States. Later, Vip3 proteins were pyramided with Cry genes (Cry1Ab + Vip3Aa20) and (Cry1Ab, Vip3Aa20, and mCry3A) in corn.

Other than commercialized crops, *Vip3* genes are successfully transformed and the proteins have been expressed in transgenic crops in laboratory-based studies. Transgenic sugar cane lines expressing Vip3A showed a high mortality rate against sugar borer *Chilo infuscatellus* (Lepidoptera: Pyraloidea) [92]. Cow pea is an important food crop in many African countries, and is harmed by Lepidopteron pest, *Maruca vitrata* (Family: Crambidae). The *Vip3Ba1* gene, isolated from Australian Bt isolates, was transformed and expressed in cowpea to provide protection against legume pod borer (*M. vitrata*), by strongly inhibiting larvae growth [93]. Similarly, a transgenic corn event (C008 and C010) expressing Vip3Aa19 has been found to be highly toxic against black cut worm [94]. When a tobacco plant was transformed with the N-terminal deletion mutant of the Vip3BR protein (Ndv200), it acquired full protection against *S. littoralis*, *A. ipsilon,* and *H. armigera* [95]. These results are helpful for future Bt-derived mutant protein transformation in crops.

## 9. Biotechnological Strategies to Improve Toxicity and Insecticidal Spectrum of Vip3 Proteins

In vitro directed evolution to increase the insecticidal potential of Vip proteins can be employed with success. The fact that Vip3 toxins share no sequence homology with Cry toxins makes them an ideal candidate for insect resistance management (IRM) programs. Vip3 proteins are found to be toxic against insects that are less susceptible to Cry toxins, such as Lepidopteron [10]. Various strategies can be employed to increase the toxicity and insecticidal spectrum of Vip proteins. For example, sequence or domain swapping to form chimeric Vip toxins successfully enhances toxicity against susceptible and resistant insects.

### Genetic Engineering of Vip3A Genes to Form Chimeric Proteins

To increase the insecticidal spectrum and activity of the Vip3 proteins, these were genetically modified. In this process, the genes were swapped for the construction of protein chimeras, where one protein expresses the sequence or domains of another protein. These novel chimeras have shown great toxicity against resistant and non-susceptible insect pests. The successful implication of a domain swapping method to generate chimeric Cry1 proteins (Cry1Ba/Cry1Ia hybrid) with enhanced toxicity against Colorado potato beetle has already been reported [96]. However, fewer reports on Vip3 chimera formation are available. A chimera of two Vip3 proteins, Vip3AcAa generated by sequence shuffling of Vip3Aa and Vip3Ac, not only had enhanced toxicity against the fall armyworm, but also to European corn borer, against which Vip3Ac was not toxic, even at high concentrations. These new chimeric proteins caused growth retardation in a Vip3A non-susceptible insect *O. nubilalis* [97]. 

Using domain shuffling, six chimeric proteins were generated by joining fragments of N-terminal, C-terminal, and the central part of the core protein from Vip3Aa and Vip3Ca. Two of these chimeras, in which only Nt domain (Vip3C having Nt domain of Vip3Aa) was shuffled, had shown no effect over stability and solubility. The exchange of the Ct domain in four chimeric proteins resulted in proteins that were insoluble and unstable to trypsin, except for one soluble and stable chimera. Compared with the parental proteins, one chimeric protein formed by shuffling the Nt domain of Vip3C with Vip3Aa showed an enhanced insecticidal activity against *S. frugiperda* (Table 2). Another chimeric protein was highly unstable and formed after shuffling Ct domain of Vip3Aa with Vip3Ca *O. furnacalis*, which is susceptible to Vip3C and non-susceptible to Vip3Aa, and also showed vulnerability against those chimeras containing the Ct domain of Vip3C. Considering the above mentioned observations, it is evident that the Ct domain is involved in the specificity of Vip3 proteins for targeted insects [98]. 

The administration of a dual toxin with non-homologous mechanisms of action is an effective way to circumvent resistance. For this purpose, chimeric proteins were formed by fusing *Vip* and *Cry* genes sequences. By combining the sequence of the full-length Vip3Aa16 toxin gene with the Nt region of the Cry1Ac activated core, a Vip3A16-Cry1Ac chimeric protein of 150 kDa was generated. The resulting fusion protein toxicity was triggered against the first-instar larva of *Ephestia kuehniella* (Lepidoptera: Pyralidae) in contrast with parental toxin Vip3Aa [33]. Similarly, another successful chimeric protein was made by fusing the nucleotide sequence of Vip3Aa7 and the Nt region of a synthetic toxin Cry9Ca, with an enhanced insecticidal activity (compared with the single parent proteins or a mixture of both) against *P. xylostella* [41]. 

Even if domain shuffling and sequence swapping are successfully implemented to form new toxin combination with improved insecticidal activity against new or resistant pests, site directed mutations and in silico analyses still provide crucial information necessary to understand protein toxicity. With the help of ever-evolving bioinformatics, it will be possible to better understand the effect of mutations on the mechanism of action of particular toxins against target insects (Table 2).

## 10. In Silico Analyses for Generation of Mutagenic Vip3 Proteins

Over the past two decades, researchers have been trying to produce mutagenic proteins with an enhanced insecticidal activity against specific pests. By using computational methodology the analysis of the chimeric protein, Vip3Aa-Cry1Ac (formed by the fusion of the functional regions of Cry1Ac and Vip3Aa), unveiled its enhanced toxicity and broad-spectrum insect control. Molecular docking analysis was performed with five Lepidopteron insect receptors, forming a strong interaction. This new protein is proposed to be the potential toxin for future crop protection against Lepidopteron pests [101]. 

### Effect of Amino Acid Modifications on Toxicity of Mutant Vip3A

Mutagenic analyses have been widely utilized to explore the amino acids present at specific sites critical for toxicity. For this purpose, Vip3A11 mutants were generated by replacing nine residues at N-terminus with Vip3A39 residues, using site targeted mutagenesis. An approximately two-fold increase in toxicity for three mutants (S9N, S193T, and S194L) was seen against *H. armigera* larvae compared with Vip3A11. Furthermore, the N-terminal amino acids also played a great role in toxicity and insect specificity against Lepidopteron pests [102]. 

Similarly, the docking and binding site prediction analysis identified the amino acids Y616, H618, Y619, W552, K557, E627, and Q652 in the Ct region as crucial sites for Vip3Aa toxin binding and insecticidal activity. The insecticidal activity of only one mutant, Y619A, was increased against *H. armigera* and *S. exigua* [38]. In another case, the Vip3Aa protein substitutions at site S164 with alanine or proline resulted in a loss of oligomer formation, and an ablation of the insecticidal potential against *S. litura*. Notably, substitution with threonine resulted in only a 35% reduction in toxicity [58].

A cysteine residue at the C-terminal region, CYS784, is a crucial site for trypsin cleavage and for the formation of the active core for toxicity. Hence, both the C- and N-terminal regions are necessary elements for toxicity. Cysteine to serine substitutions at the C-terminal also reduced the Vip3A7 protein insecticidal activity against *P. xylostella,* likely due to the disruption of the disulfide bonds between the cysteine residues [41]. A modified Vip3Ca protein, ARP150v02, with amendments at eight locations near the N-terminus region, was expressed and purified in *E. coli*. The ARP150v02 protein showed an insecticidal activity against many insects, but a high insecticidal effect against *S. frugiperda* (LC_50_ = 450 ng/cm^2^). In contrast, this protein was ineffective against *H. armigera,* even at a high dose. The binding assays revealed that the ARP150v02 protein competes for binding with Vip3Aa in *S. frugiperda* [103]. More studies based on site directed mutagenesis are necessary in order to overcome pest resistance. 

## 11. Synergism and Antagonism in Vip3 and Cry Proteins 

Various studies reported the presence of synergism in Vip3 and other Bt toxin (Cry and Cyt). The coexpression of Vip3Aa and Cyt2Aa in *E.coli* lead to synergism in *S. exigua* and *Chilo suppressalis* (Lepidoptera: Crambidae) [104]. Similar synergism was observed between Vip3A and Cry1Ia in *S. frugiperda*, *Spodoptera albula*, and *Spodoptera cosmioides* (Lepidoptera: Noctuidae) [105]. Another synergistic combination was found between Cry9Aa and Vip3Aa, possibly due to the binding mechanism between two toxins with BBMVs in *C. suppressalis* and *O. furnacalis.* Interestingly, the synergism between Vip3Aa and Cry9Aa mutants was disturbed moderately in *C. suppressalis,* and severely in and *O. furnacalis.* Synergism resulted in an improved toxicity of Vip3Aa and Cry9Aa in *C. suppressalis,* which is a great threat to rice crops in China [106]. Strong synergism was also seen in the Cry1Ab/Vip3Ca protein combination. These can be useful in *M. separate* and *O. furnacalis* control in future pyramided gene stacking [72]. A high rate of synergism was identified in the Vip3Aa and Cry1Ab combination against the neonatal larvae of *S. frugiperda,* without competing for binding sites [107]. In the same study, several other Bt protein combinations, like Vip3Aa/Cry2Ab, Cry1Ab/Cry2Ab, Cry1Ab/Cry2Ab/Vip3Aa, Cry1Ea/Cry1Ca, and Vip3Ca/Cry1Ea, also showed synergism against *S. frugiperda*. 

Finally, it is worth mentioning that some toxin combinations show antagonism as well. For example, Vip3Aa showed slight antagonism with Cyt2Aa in *Culex quinquefasciatus* (Diptera: Culicidae) [104], and with Cry1Ia in *Spodoptera eridania* (Lepidoptera: Noctuidae) [105]. These antagonisms could result from direct competition between the CRY and Vip3A toxin for the same binding sites in insect BBMVs [108]. Another study identified many antagonist combinations in various Cry and Vip3 protein pairs, such as Vip3A/Cry1A or Cry1Ca, Cry1Ca/Vip3Aa, Cry1Ca/Vip3Ae, Cry1Ca/Vip3Af, Vip3Af/Cry1Aa, or Cry1A. Vip3A and Cry1Ca showed more antagonism in *S. frugiperda* at LC_90_ [109]. This knowledge can be helpful in the future for stacking genes in pyramids in order to broaden the insect spectrum, and for managing the evolution of insect resistance.

### Efficacy of Pyramided Vip3 and Cry Proteins 

The past decade marked an increased use of Vip3A with Cry proteins in pyramided crops for broader insecticidal activity and in insect resistance management [78]. The pyramiding of Cry1A and Cry2A with Vip3A is a promising strategy in IRM programs. No cases of cross resistance have been reported yet in pyramided Bt crops. Meanwhile, registered varieties of pyramided Bt cotton and maize containing Vip3Aa19 and Vip3Aa20 are commercialized worldwide [110]. 

In order to form pyramided Bt rice, a fusion gene (C1V3) was formed by combining truncated Cry1Ab and the full-length Vip3A by a linker, to generate a chimeric protein that could be digested efficiently by trypsin. After digestion into activated fragments, both toxins function just like an individual toxin of Cry1Ab and Vip3A. Transgenic rice with the fusion gene (C1V3) showed insecticidal activity against two major rice pests, *C. suppressalis* and *Cnaphalocrocis medinalis* (Lepidoptera: Crambidae). A high toxin content was seen after two generations in fields, along with disease spots. No difference in phenotypes was seen in the transgenic (A1L3) and control rice plants. Further investigations will clarify the implications of this strategy [111]. Cry1Ac and Vip3Aa are potential candidates for sugarcane protection against *Diatraea flavipennella* (Lepidoptera: Crambidae) and *Elasmopalpus lignosellus* (Lepidoptera: Pyralidae), through pyramided transgenic Bt sugarcane production and commercialization [112]. 

The main threat to the commercialization of these crops is the existence of resistance to Cry1A or Cry2A, which may spark the evolution of resistance to these pyramided crops [110]. To overcome this barrier, other strategies could be implemented, alone or by pyramiding them with Bt toxins, in order to control resistant pests. For this purpose, post transcriptional gene silencing of insects specific genes involved in various physiological functions could be effective to inhibit insect growth and development. In this method, double stranded RNAs (dsRNA) are designed to target essential insect genes, disrupting their expression by RNA interference (RNAi). Short sequences of dsRNA are incorporated into insects through diet and are also transformed in plants [113,114]. The chances of cross resistance are very low in this case, as both pathways have diverse and independent mechanisms of action.

Pyramided Cry toxins and RNAi corn plants targeting *D. v. virgifera* have already been developed [115]. Another pyramid formed by the combination of Bt toxins, Cry1Ac and dsRNA, designed to target the metabolism of juvenile hormone (JH) in *H. armigera*, was introduced in cotton. Two types of cotton plants, JHA (targeting JH acid methyltransferase) and JHB (JH transporter protein), showed a high activity against resistant *H. armigera* [116]. For the safety and high efficiency of this strategy, dsRNA can be transformed into plastids to be expressed with plastid genomes rather than a nuclear genome. The chloroplast transformation of dsRNA also increases the protein content in the cell, as RNAi machinery is absent in the chloroplast compartment. Introducing dsRNA into potato plastids targeted the β-actin gene of the deadly potato pest, Colorado potato beetle, and protected the crop against this notorious pest [117] (Table 3).

## 12. Future Perspectives 

There is an increasing need for new IRM strategies. The effective control of resistant pests and the delay of adaptive evolution to resistance in insects will not be achieved solely with pyramid strategies. In depth knowledge of the mechanism of action of Bt proteins and the mechanisms of insect resistance is crucial for prolonged benefits of Bt toxins in pest control. For effective resistance management, it is necessary to develop novel toxins, and to combine more than one strategy in pest control. Vip toxins are a promising new generation of insecticides to be used in spray formulation and transgenic crops, because of their broad spectrum of insect targets. Researchers are focusing on the structure and function of Vip proteins and are attempting to find new Vip proteins from already identified and novel Bt strains. Finding new proteins could be a strategy of choice to manage resistance to Bt toxins. Next generation sequencing (NGS) can accelerate the discovery of novel proteins through the complete sequencing of novel genomes and already known Bt strains. 

## Figures and Tables

**Figure 1 toxins-12-00522-f001:**
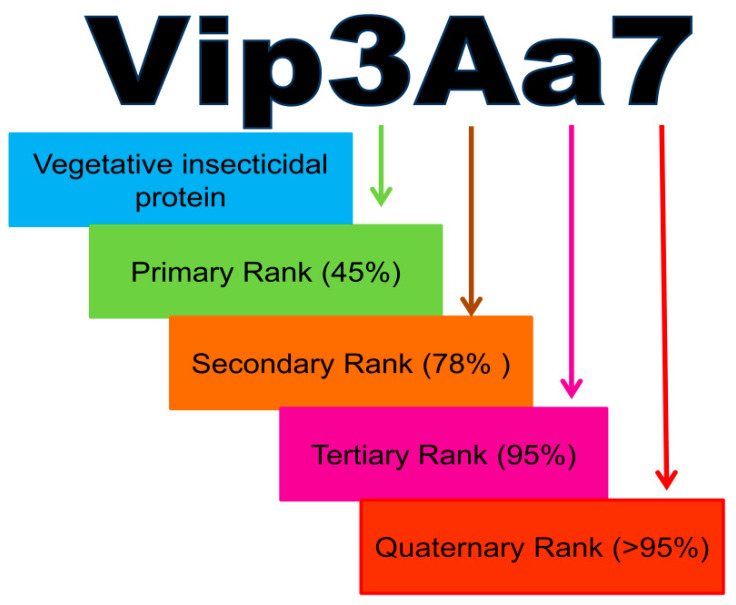
Schematic representation of vegetative insecticidal proteins (Vip) proteins’ nomenclature system. Each protein is assigned a four ranked name—primary rank, given to proteins sharing less than 45% homology in amino acid sequences; secondary and tertiary ranks, with less than 78% and 95% similarity, respectively; and finally, the quaternary rank is given with more than 95% identical proteins.

**Figure 2 toxins-12-00522-f002:**
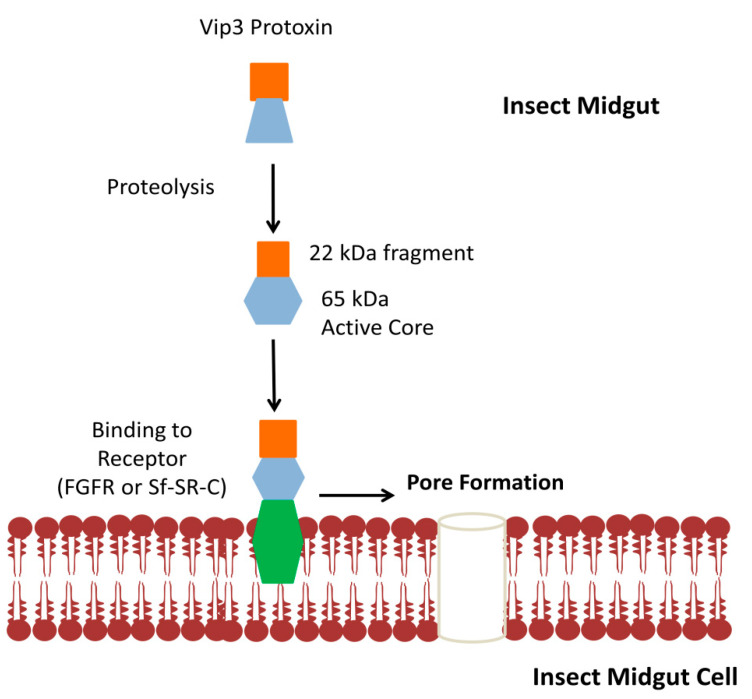
Proposed mechanism of pore formation by the Vip3 toxin. The Vip3A toxin is activated by proteolysis inside the insect midgut. In the next step, activated toxins, including 22 kDa and 65 kDa fragments, bind with receptors, leading to pore formation in the insect midgut cells and, ultimately, to the death of the insects.

**Figure 3 toxins-12-00522-f003:**
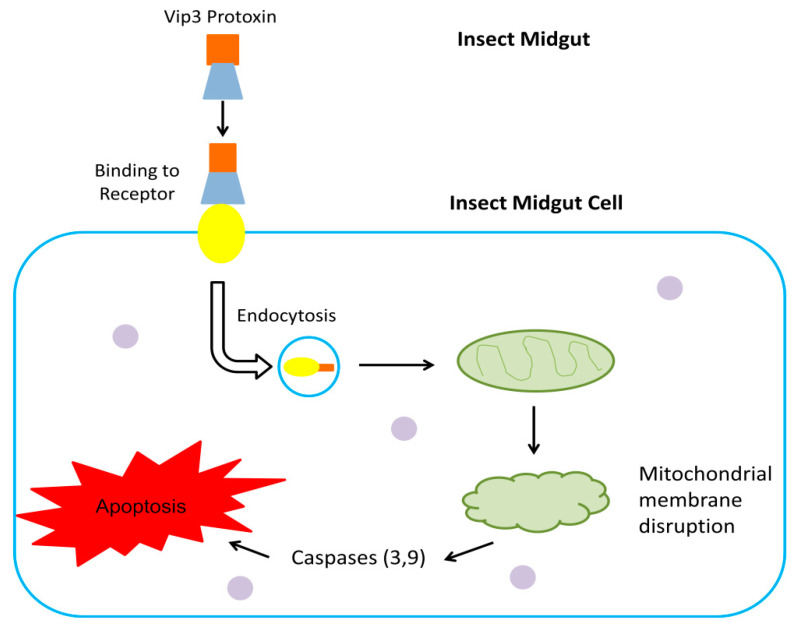
Schematic representation of the mechanism of Vip3 toxin induced apoptotic cell death of insect midgut cells. Vip3A protoxin binds with receptors, and the receptor mediated internalization of toxin takes place. Toxin internalization leads to changes like DNA damage, mitochondrial membrane disruption, and the activation of caspases (caspase 3 or 9), in turn promoting apoptotic cell death.

**Table 1 toxins-12-00522-t001:** Comparison of Vip family proteins.

Traits	Vip1/Vip2	Vip3	Vip4
Number of proteins	15 Vip1 20 Vip2	111	5
Gene size	~4 kb	~2.4 kb	2895 bp
Number of amino acids	881 Vip1 462 Vip2	787 to 789	965
Protein size	Vip1 80 kDa Vip2 45 kDa	89 kDa	~108 kDa
Target insects	Coleopteran and Hemipteran	Lepidopteron	Not available
Mode of action	ADP ribosyltransferase Activity/cytoskeleton abnormalities	Apoptotic cell death/pore formation	Not available
Commercialized crops	None	Cotton and maize	None

**Table 2 toxins-12-00522-t002:** Chimeric Vip3 proteins and their toxicity profiles.

Toxin	Chimera Type ^1^	Insecticidal Activity ^2^	Reference
**Vip3AcAa**	Chimera of Vip3Ac N terminus + Vip3Aa C-terminus	Toxic to *O. nubilalis*, insecticidal activity against *S. frugiperda*, *H. zea*, and *Bombyx mori*	[97]
**Vip3AaAc**	Chimera of Vip3Aa N-terminus + Vip3Ac C-terminus	RA against *S. frugiperda* and *H. zea*; LA against *B. mori*	
**Vip3Aa14**	Chimera of Vip3Aa14 and Cry1Ac	Toxic against *H. armigera* and *P. xylostella*; RA against *S. litura* than Vip3Aa	[99]
**Vip3Aa7**	Chimera of Cry1C promoter + Vip3Aa7 (Nt 39 aa deleted) + Cry1C C-terminus	RA against *P. xylostella*, *H. armigera*, and *S. exigua*	[100]
**Vip3Aa7**	Chimera of Vip3Aa7 + Cry9Ca N-terminus	Increased activity against *P. xylostella*	[41]
**Vip3Aa16**	Chimera of Vip3Aa16 + Cry1Ac N-terminus (48-609 aa)	Increased toxicity against *E. kuehniella*	[33]
**Vip3Aa and Vip3Ca**	Chimera of Vip3Aa N-terminus + Vip3Aa central domain + Vip3Ca C-terminus	Not active against *A. gemmatalis*, *M*. *brassicae*, *O. furnacalis*, *S. frugiperda*, *S. littoralis*, *H. armigera*, and *S. exigua*	[98]
	Chimera of Vip3Aa N-terminus + Vip3Ca central domain + Vip3Ca C-terminus	RA against *Anticarsia gemmatalis*, *M. brassicae*, and *O. furnacalis*; strong activity against *S. frugiperda* than Vip3Ca	
	Chimera of Vip3Aa N-terminus + central domain of Vip3Ca + Vip3Aa C-terminus	Insoluble protein	
	Chimera of Vip3Ca N-terminus + central domain of Vip3Aa + Vip3Ca C-terminus	LA against *S. littoralis*, *S. frugiperda*, *H. armigera*, *M. brassicae*, *S. exigua,* and *A. gemmatalis;* except *O. furnacalis*	
	Chimera of Vip3Ca N-terminus + central domain of Vip3Ca + Vip3Aa C-terminus	Insoluble protein	
	Chimera of Vip3Ca N-terminus + central domain of Vip3Aa + Vip3Aa C-terminus	RA against *S. littoralis*, *S. frugiperda*, *H. armigera*, and *M. brassicae*; except *S. exigua* compared with Vip3Aa. LA against *A. gemmatalis* and *O. furnacalis*	

^1^ aa = amino acids; ^2^ LA = lost activity; RA = reduced activity.

**Table 3 toxins-12-00522-t003:** Commercialized pyramided Bt crops with the Vip3A protein [118,119].

Plant	Event	Pyramid	Target Insects	Country
**Maize**	BT11/GA21	Cry1Ab, Vip3Aa20	*A. ipsilon*, *O. nubilalis*, *H. zea*, and *S. frugiperda*	Canada (2005), South Korea (2006/2008), Japan, Mexico, Philippines (2007), Argentina, Brazil (2009), Uruguay (2011), and Colombia (2012)
	BTT11/GA21/MIR162	Cry1Ab, Vip3Aa20	*H. zea*, *S. frugiperda*, and *A. ipsilon*	Brazil (2011) and Colombia (2012)
	BT11/MR162	Cry1Ab, Vip3Aa20	*A. ipsilon*, *O. nubilalis*, *H. zea*, *S. frugiperda*, and *Spodoptera albicosta*	United States (2009)
	BT11/MIR162/MIR604	Cry1Ab, mCry3A, Vip3Aa20	*O. nubilalis*, *Diatraea crambidoides*, *S. frugiperda*, *Pseudaletia unipunctata*, *S. exigua*, *A. ipsilon*, *Striacosta albicosta*, *Diatraea saccharalis*, *Diabrotica virgifera*, *Diabrotica barberi*, and *Papaipema nebris*	United States (2009)
	BT11/MIR162/MIR604/GA21	Cry1Ab, mCry3A, Vip3a20	*Diabrotica spp.*, *H. zea*, *O. nubilalis*, *S. frugiperda*, and *A. ipsilon*	Colombia (2012)
	BT11 × MIR162 × TC1507 × GA21	Cry1Ab + Cry1Fa + Vip3Aa	Lepidopteran	United States (2011)
	BT11 × MIR162 × MIR604 × TC1507 × 5307	Cry1Ab + mCry3A + Vip3A + chimeric (Cry3A-Cry1Ab)	Lepidopteron and Coleopteran	Brazil (2019)
**Cotton**	COT102 × COT67B	mCry1Ab + Vip3Aa19	*H. virescens*, *H. zea*, *P. gossypiella*, *S. frugiperda*, *S. exigua*, and *T. ni*	United States (2008)
	COT102 × COT67B × MON88913	Cry1Ab + Vip3A	Lepidopteron	Costa Rica (2009)
		Cry1Ac + Cry1Fa + Vip3Aa	Lepidopteron	United States (2013)
	COT102 × MON15985	Cry1Ac + Cry2Ab + Vip3A	Lepidopteron	Japan, Australia, and Mexico (2014)
	COT102 × MON15985 × MON88913	Cry1Ac + Cry2Ab + Vip3A	Lepidopteron	Japan, Australia (2014), Brazil, Taiwan (2016), and South Korea (2015),
	COT102 (IR102)	Vip3A	Lepidopteron	Australia, New Zealand, United States (2005), Canada (2011), Costa Rica (2017), South Korea (2014), Mexico (2010), Japan (2012), Taiwan, Philippines China (2015), and Columbia (2016)
	3006-210-23 × 281-24-236 × MON88913 × COT102	Cry1Ac + Cry1Fa + Vip3Aa	Lepidopteron	Japan (2016), Mexico, and South Korea (2014)
	281-24-236 × 3006-210-23 × COT102 × 81910	Cry1Ac + Cry1Fa + Vip3Aa	Lepidopteron	Japan (2016) and Brazil (2019)
	281-24-236 × 3006-210-23 × COT102	Cry1Ac + Cry1F + Vip3A	Lepidopteron	Brazil (2018)
	T304-40 × GHB119 × COT102	Cry1Ab + Cry2Ae + Vip3A	Lepidopteron	Brazil (2018)
	GHB811 × T304-40 × GHB119 × COT102	Cry1Ae + Cry2Ae + Vip3A	Lepidopteron	Brazil (2019)
	GHB614 × T304-40 × GHB119 × COT102	Cry1Ab + Cry2Ae + VIP3A	Lepidopteron	Philippines (2020)

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
