# Peer review of "Current Insights on Vegetative Insecticidal Proteins (Vip) as Next Generation Pest Killers"

_toxins, 2020, doi:10.3390/toxins12080522_

Round 1
Reviewer 1 Report
The manuscript is well-written and logically constructed but have some overall problems. My commets is in the pdf file

Reviewer 2 Report
The presented manuscript is a good and comprehensive review dedicated to the insecticidal factor Vip of Bacillus thuringiensis bacterium. The review provide good representation of what is currently known about different families of Vip toxins. The only drawback of the manuscript is that the different families are described separately and little comparison is provided. It would be better to add the information regarding differences and similarities of Vip1/2, Vip3, Vip4 groups. Also, sections 5, 7, 8, 9, 10 are dedicated to the Vip3 only. It seems a little confusing, that no information regarding other Vip toxins are provided.
Round 2
Reviewer 1 Report
This manuscript reports an investigation secretes various 4 proteins during different growth phases with insecticidal potential against many economically 5 important crop pests. One of the important families of Bt proteins is vegetative insecticidal proteins 6 (Vip), which are secreted into the growth medium during vegetative growth.The work is sound, the presentation is excellent.
line 152 Spodoptera italics please
